# Cisplatin Mouse Models: Treatment, Toxicity and Translatability

**DOI:** 10.3390/biomedicines9101406

**Published:** 2021-10-07

**Authors:** Martina Perše

**Affiliations:** Medical Experimental Centre, Institute of Pathology, Faculty of Medicine, University of Ljubljana, 1000 Ljubljana, Slovenia; martina.perse@mf.uni-lj.si; Tel.: +386-4047-4675

**Keywords:** cisplatin, toxicity, kidney, gut, nerve system, mouse, treatment, tumors, mouse model

## Abstract

Cisplatin is one of the most widely used chemotherapeutic drugs in the treatment of a wide range of pediatric and adult malignances. However, it has various side effects which limit its use. Cisplatin mouse models are widely used in studies investigating cisplatin therapeutic and toxic effects. However, despite numerous promising results, no significant improvement in treatment outcome has been achieved in humans. There are many drawbacks in the currently used cisplatin protocols in mice. In the paper, the most characterized cisplatin protocols are summarized together with weaknesses that need to be improved in future studies, including hydration and supportive care. As demonstrated, mice respond to cisplatin treatment in similar ways to humans. The paper thus aims to illustrate the complexity of cisplatin side effects (nephrotoxicity, gastrointestinal toxicity, neurotoxicity, ototoxicity and myelotoxicity) and the interconnectedness and interdependence of pathomechanisms among tissues and organs in a dose- and time-dependent manner. The paper offers knowledge that can help design future studies more efficiently and interpret study outcomes more critically. If we want to understand molecular mechanisms and find therapeutic agents that would have a potential benefit in clinics, we need to change our approach and start to treat animals as patients and not as tools.

## 1. Introduction

In 2018, 18.1 million new cases and 9.5 million cancer-related deaths were diagnosed worldwide. It is estimated that by 2040 the number of new cancer cases per year will rise to 29.5 million and the number of cancer-related deaths to 16.4 millions [1].

Despite intensive research and progress in cancer therapy, chemotherapeutic drugs are still the basis of systemic therapy for many cancers. Cisplatin is one of the most widely prescribed chemotherapeutic drugs, used to treat a wide range of pediatric and adult malignances such as ovarian, testicular, bladder, head, neck, breast and lung [2,3]. It is prescribed in nearly 50% of all tumor chemotherapies [4]. However, it has limited use in clinical practice due to various deleterious side effects. Currently, around 40 side effects of cisplatin have been reported [5]. Extensive supportive medical care of cisplatin treated cancer patients enables the use of very high-dose cisplatin regimens [3,6,7,8]. However, with the use of high-dose treatment regimens acute kidney injury, persistent diarrhea, neurological disorders and loss of hearing became major hurdles of cisplatin therapy. These unwanted effects result in reduction or cessation of therapy or have a major impact on patients’ quality of life, leading to higher levels of negative states such as depression and anxiety. There is no effective therapy for the prevention of these side effects; the current treatment strategy is symptomatic with limited effectiveness.

Based on extensive research, and after 40 years of cisplatin use, the anti-cancer effects of cisplatin are well understood, while the mechanisms of cisplatin toxicity remain unclear [9,10,11]. Therefore, increasing emphasis is being placed on various strategies to reveal the mechanisms responsible for toxicities and to overcome cisplatin side effects. Cisplatin mouse models are a promising strategy; however, despite intensive investigation and numerous promising results, no significant improvement in the treatment outcomes has been reached in clinical practice [12,13,14].

The aim of the present paper is firstly to illustrate the complexity of cisplatin mouse models. The review summarizes the data to demonstrating that mice respond to cisplatin treatment in a similar way as humans. Mice develop all the same cisplatin side effects that humans do. However, in contrast to cisplatin treated cancer patients, in which all cisplatin side effects are monitored and treated, in animal studies usually one cisplatin toxicity is under investigation, while other side effects of cisplatin are mostly neglected or ignored. For instance, in cisplatin nephrotoxicity or neurotoxicity studies gastrointestinal injury is usually neglected. Secondly, to encourage researchers to take into consideration all events that are taking place in a mouse and to reconsider the severity and the time course of toxicity in accordance with other interdependent and interconnected mechanisms and toxicities in the body. Understanding the complexity of cisplatin side effects in a dose- and time-dependent manner as well as the interconnectedness and interplay of pathomechanisms among tissues and organs can help design future studies more efficiently and interpret study outcomes more critically. Thirdly, the review aims to expose limitations and weaknesses of current cisplatin protocols together with suggestions for future studies. It is important to recognize that not only the lack of complex knowledge and approaches but also a lack of robust and validated cisplatin mouse models are important factors that contributed to poor translatability. Since the literature is extremely numerous, only the most relevant articles are included.

## 2. Cisplatin Mouse Models

Cisplatin mouse models have been used to investigate pharmacokinetics and tissue distribution of cisplatin [15,16,17], the repair capacity of cisplatin-DNA adducts [18], the molecular mechanisms of cisplatin toxicity [19,20,21,22,23] and to test a new generation of platinum-based chemotherapy drugs or adjunctive therapies [24,25,26,27,28,29,30,31,32,33,34,35], or other potential agents or strategies to prevent or treat cisplatin toxicities [36,37,38,39,40,41].

Table 1 shows cisplatin protocols in mice reported in publications in 2020/2021. Nephrotoxicity [42,43,44] was by far the most frequently studied toxicity, followed by neurotoxicity [45,46,47], ototoxicity [16,48,49], gonadotoxicity [50,51,52,53], gastrointestinal toxicity [54,55,56], muscle wasting [57,58,59] and anemia [60].

A MEDILINE/PubMed search, using keywords “cisplatin”,“mouse”,“toxicity” was conducted in February 2021. Due to a huge number of publications (more than 10.000 results), the search was limited to studies from April 2020 to February 2021. One hundred full text articles were retrieved and examined. Special attention was paid to the cisplatin treatment protocol, duration of the study, model (healthy or tumor bearing), humane intervention and endpoints used, clinical markers of toxicity, necropsy or histology findings, randomization (group allocation). To better evaluate cisplatin treatment protocols in mouse studies, publications were divided in two groups. In the first group, cisplatin treatment was used mostly as a positive control to evaluate the antitumor activity of a novel agent or treatment strategy in tumor bearing mice. Due to huge variability in cisplatin protocols, results are presented in Appendix A. In the second group cisplatin treatment was used to investigate cisplatin toxicity in healthy mice. Most frequently used cisplatin protocols for specific toxicity are presented in Table 1. In toxicity studies only the toxicity under investigation was examined, while other side effects of cisplatin were not reported. Hydration was rarely used, supportive care never. Legend: ip—intraperitonealy, d—day.

## 3. Cisplatin Nephrotoxicity

Experimental nephrotoxicity, first reported in 1971 [61], is the most frequently studied cisplatin side effect. Over the past decades, researchers have demonstrated that cisplatin can cause nephrotoxicity or acute kidney injury (AKI) of varying severity in a dose-dependent manner [62]. Depending on the dose (single or cumulative) rodents may develop acute (early) or chronic (advanced) kidney injury. However, AKI evolves slowly and predictably after initial and repeated exposure. Unlike other drug toxicities, clinical evidence of cisplatin nephrotoxicity develops within a few days after administration. In clinical practice, nephrotoxicity typically presents approximately 10 days after cisplatin treatment [3]. In mice, clinical evidence of nephrotoxicity develops 4–6 days after a single sub-lethal dose of cisplatin (Figure 1). To better understand development of the nephrotoxicity after a single sub-lethal nephrotoxic dose of cisplatin, the development over time of morphological, functional, and clinical changes is schematically presented in Figure 1.

In patients, cisplatin treatment is usually administered in cycles with 1- or 3-week intervals and a cycle consisting of a single high dose of cisplatin or multiple lower daily doses (see Table 2). In mice or rats, nephrotoxicity is mostly induced by a single cisplatin administration. Repeated cisplatin protocols for nephrotoxicity are extremely rare [44], which is also the main critique of AKI models [13]. Moreover, in mice, a wide variation of cisplatin dosage is used to induce renal toxicity, i.e., from low sub-therapeutic (5 mg/kg), sub-lethal nephrotoxic (10–12 mg/kg) to lethal dosage (14–18 mg/kg) or even higher (>20 mg/kg) [44]. The use of a different dosage of cisplatin can be useful when the time course and/or the severity of nephrotoxicity and its functional, morphological or molecular abnormalities are under systematic investigation. However, for testing potential agents or treatment strategies we need a robust and validated cisplatin mouse model. Currently, there is no standardized, robust or validated cisplatin mouse model of AKI that is clinically or physiologically relevant to patients [44].

Recently, Siskind and coworkers [75] established a mouse model of repeated administration of cisplatin (FVD, 7 mg/kg per week for 4 weeks) [76]. However, their aim was to obtain a model for chronic kidney disease [76,77,78]. In the past, it has already been demonstrated that cisplatin can have long term effects on kidney morphology and function after single [64,79,80] or repeated [81,82,83] cisplatin administration. However, the long-term toxic effects of cisplatin became the subject of investigation recently, when it was realized that even a mild and reversible AKI can have long term effects in patients [84,85,86] or that chronic kidney disease may develop undetected [87,88].

### 3.1. Pathopysiological Mechanisms

The severity of kidney injury (mild, moderate, severe) and consequently functional, morphological, molecular, and inflammatory alterations in the kidney as well as morbidity and mortality depends on the cisplatin dosage and the time of study termination [44]. The pathophysiological mechanisms of cisplatin AKI in rodents involve cellular uptake, damage of proteins, lipids and mitochondria, oxidative stress, disruption of the cytoskeletal integrity of the cell polarity, alterations in membrane proteins and water channels, leading to damage of epithelial cells of renal tubules, loss of brush border, activation of cytokines, receptors and inflammatory cells, and finally reduced reabsorptive capacity, which reflects clinically as polyuria, proteinuria, glycosuria, electrolyte wasting (hyponatremia, hypomagnesemia, hypokalemia, hypocalcemia), reduced creatinine (Cr) clearance and glomerular filtration rate (GFR) and failure to clear nitrogenous wastes from the blood. As a result, blood urea nitrogen (BUN) and uric acid accumulate in the blood (described in detail in [44]). Intensive investigation of molecular mechanisms of cisplatin nephrotoxicity resulted in a plethora of information, including contradictory ones. The latter is the consequence of above-mentioned heterogeneity of cisplatin protocols. Namely, the course and the signature of underlying mechanisms (i.e., severity of oxidative stress, intensity of inflammation, activation of particular immune cell types, inflammatory and molecular crosstalk and response, type of cell death, etc.) strongly depend on the severity of AKI (mild, moderate–reversible, severe, irreversible) or cisplatin dose (subtherapeutic, therapeutic, lethal, intoxication). An update on molecular mechanisms involved in the pathogenesis of cisplatin nephrotoxicity can be found elsewhere [93,94,95,96].

### 3.2. Weaknesses and Translatability

To study AKI, researchers use a single dose of cisplatin in a dosage above LD100 and terminate study 48–96 h after cisplatin treatment (Table 1). To avoid the inevitable death of animals, they use shorter endpoints. Differences in AKI severity are then confirmed by the histology report, BUN, Cr or other molecular markers. If differences in AKI severity are significant, the testing agent or strategy is evaluated as beneficial and promising [37,44]. The problems of such studies are numerous. First, the lethal dosage of cisplatin causes systemic toxicity and multi-organ failure, which clinically represent different pathology (i.e., intoxication) and treatment (i.e., detoxication). Second, evaluation of potential treatment strategies based on the significant differences in BUN, Cr, renal histology or survival in such protocols does not have any clinical useful value. At the end, animals die despite significant improvements in some markers (i.e., BUN, Cr, severity of tubular necrosis, time of mortality), which is very likely an effect of biological variability (inter-individual variability is high in both, mice and humans). In humans, severity of AKI is graded according to the levels of serum creatinine: grade 1: increased serum Cr levels to 1.5–1.9 times baseline, grade 2: increased Cr levels to 2.0–2.9 times baseline, grade 3: increased Cr levels to 3.0 times baseline or >4.0 mg/dl or initiation of renal replacement therapy [12]. Third, at autopsy and when interpreting results scientists often forget or ignore other pathologies or side effects (i.e., gastrointestinal, myelotoxicity, anemia, vasculitis, etc,) which significantly affect the study outcomes and hamper comparison of results and development of valid therapeutic strategies. Forth, cisplatin treated mice usually do not receive any supportive care. Moreover, some studies even use water deprivation prior cisplatin treatment [44]. Hydration and supportive care affect nephrotoxicity and mortality enormously and also influence Cr and BUN levels. Dehydration, degradation due to starvation or loss of body weight, gastric or intestinal bleeding [97,98,99], all of which are usually seen in cisplatin models (see gastrointestinal toxicity section) affect the levels of BUN/Cr resulting in misinterpretation of the actual degree of renal damage [65,97,99].

In humans, treatment with cisplatin consists of repeated cycles of as high a dose as possible. The dose in humans is balanced between antitumor efficacy and toxicity to avoid unacceptable toxic side effects. However, the patient is constantly monitored and provided with extensive supportive medical care (i.e intravenous hydration, diuretics, slow infusion of the drug, anti-emetics) [3,12,100]. Despite all care measures, severe kidney injury is observed in one third of cisplatin treated patients (inter-individual variability; Table 2) [3,12]. In addition, 16–40% of the patients treated with cisplatin develop myelotoxicities, of which leukocytopenia and neutropenia have the highest incidence [89]. Depending on a dose (single or cumulative) cisplatin can cause leucocytopenia, thrombocytopenia, and anemia also in mice (toxicity on hemopoietic cells, Figure 1). Myelotoxicity can be observed already a day after cisplatin administration and is more toxic for earlier hemopoietic progenitor cells than for the mature cells [74].

To improve cisplatin protocols, it is thus important to understand why selection of certain cisplatin dosage, the time-point of measurement or observation (i.e., scientific endpoints) and the use of supportive care are the key variables that directly affect the measured outcomes of a study and the translatability. Not only the severity of cisplatin nephrotoxicity but also the incidence of nephro-, myelo-, neuro-, oto-toxicity are dose-related in both, humans and animals (explained in the following sections). The dose of cisplatin (single and/or cumulative) is thus important not only from the animal welfare point of view but mostly from the scientific and clinical point of view. When evaluating potential treatment agents both, sub-therapeutic and lethal dosages result in a lack of translatability, unnecessary suffering of animals and time and money costs.

### 3.3. Mouse Equivalent Dose–Simplistic Pharmacological Guides

Some research papers refer to mouse cisplatin dose, which was calculated from the human clinical dose using simplistic calculation. Table 2 includes examples of such calculations. Based on numerous studies investigating maximum tolerated dose (MTD) or lethal dose (LD100) of cisplatin in mice (Table 3) it is obvious that such simplistic guides for dose conversion between animals and humans [92] (or other similar papers) can do more harm than benefit. “This overly simplistic conversion neglects discussion of interspecies differences in drug absorbance, metabolism, clearance, etc. These differences in pharmacokinetics greatly affect the resulting peak plasma concentration (C_max_) values and exposure derived from area under curve (AUC), which influence the dose response relationship of potential therapeutics” [101]. In addition, in humans cisplatin is given as intravenous 1–2 h infusion treatment with pre and post hydration with up to 3 L of saline, while in mice intraperitoneal administration without any hydration is usually used. Both, route of administration and particularly hydration have profound effects on the distribution and elimination rates and consequently LD100 or MTD [91]. LD100 and MTD doses for intraperitoneal administration of cisplatin in mice without hydration are summarized in Table 3. However, it is important to emphasize that currently used MTD dose in mice does not necessary represent a clinically relevant dose of cisplatin for therapeutic efficacy [15]. Concentration may vary between a MTD dose in mice and concentration achieved in humans (due to above explained reasons) which means that study outcomes can have limited value from translational perspective of a drug. Since doses of cisplatin in published animal studies vary widely (from 1–40mg/kg, Table 1, Appendix A) a publication with a clinically relevant cisplatin doses in mice, like published for some other drugs [102], is more than needed.

As explained above, the dose (single/cumulative) and the timepoint are two variables that directly relate to severity and the incidence of cisplatin nephrotoxicity. However, in the following sections, it will be shown that cisplatin induces also gastrointestinal toxicities in the body whose severity is also dose dependent and can affect cisplatin nephrotoxicity significantly. The kidney participates in the control of fluid osmolality, acid-base balance and electrolyte concentrations (i.e., Mg, K, Na, Cl), and is the main organ responsible for filtration and detoxification of the blood and is thus directly confronted with all toxins, cytokines, detrimental waste products, or microorganisms that are flushed or penetrate into the circulation from or though other organs.

## 4. Cisplatin Gastrointestinal Toxicity

Cisplatin is one of the most emetogenic drugs in the clinic [107] causing profound and long lasting gastrointestinal symptoms such as nausea, vomiting, bloating, diarrhea, constipation [108,109]. Gastrointestinal side effects can occur in up to 40% patients receiving standard dose chemotherapy or 100% patients receiving high dose chemotherapy. Gastrointestinal problems can persist up to 10 years after the treatment cessation (late/chronic toxicity). Despite guidelines to navigate management of gastrointestinal side effects, diarrhea is responsible for about 5% of early deaths during chemotherapy [110].

Cisplatin can cause acute (within 24 h) and delayed vomiting/pica (24 h after cisplatin) in both, humans and rodents [111,112,113]. In rodents, acute vomiting reflects as a reduction in food intake, an increase in non-nutritive substance intake, and a delay in gastric emptying (so-called pica behavior; rodents do not have vomiting reflex). Acute pica occurs after low and high cisplatin doses, while delayed pica, including gastric stasis and stomach distension is dose-dependent (single or cumulative) [56,112,113,114] and worsens after repeated cisplatin administration [113]. 

Cisplatin causes damage to the gastrointestinal mucosa along the whole gastrointestinal tract (the stomach, small intestine and colon), however in the colon mucosal lesions appear later and are less severe [72]. Alterations are seen in the morphology [72], kinetics, secretory and digestive function and nutrition uptake [108,115]. Changes are similar to those observed in humans. Mucosal damage after single cisplatin injection can persist up to 10 days [72]. The severity of mucosal damage along the gastrointestinal tract is dose-dependent [72]. Mucosal damage with inflammation, digestive dysfunction, disruption of water and electrolyte balance are responsible for dehydration, malnutrition, and changes in feces consistency [110]. However, an initial increase in gastrointestinal transit, associated with acute intestinal inflammation, is followed by a slowing in transit. Recent studies have shown that cisplatin can cause morphological and functional alterations in the enteric neurons in a dose-dependent manner [116]. Partial loss of enteric neurons and gial cells [55] was suggested to be responsible for reduced gut motility (Figure 2). 

### 4.1. Weaknesses and Translatability

Like humans also cisplatin treated mice suffer from nausea/pica, stomach and gut inflammation, abdominal pain, and have reduced food and water intake and altered feces consistency (from sticky, loose to diarrhea), all of which by itself is a risk factor for kidney impairment. Therefore, extensive hydration and treatment of gastrointestinal symptoms are routinely applied into clinical settings to reduce intravascular depletion of fluid and electrolytes (Mg, K, Na, Cl) and consequently the incidence and the severity of renal injury in cisplatin treated patients [3]. An example of an incidence and severity of gastrointestinal symptoms in cisplatin treated cancer patients is shown in Table 4. However, in cancer patients, age, co-morbidities (diabetes, hypertension), and concomitant nephrotoxic medications (antibiotics-infections, NSAID, etc) can increase the risk of cisplatin-induced kidney injury [3]. Many cisplatin treated patients have thus AKI with mixed renal etiology, while in cisplatin animal studies AKI is mostly result of severe dehydration, malnutrition, electrolyte wasting, systemic toxicity, and cisplatin nephrotoxicity [117]. In addition, up to 100 % of patients develop Mg depletion, which has been associated with increased cisplatin transport to the kidney and enhanced cisplatin nephrotoxicity [3,12]. Thus, the development of AKI in mice and in cancer patients differs in the etiology, underlying mechanisms and importantly, the treatment [117].

### 4.2. Mechanisms

Investigation of the molecular mechanisms involved in gastrointestinal toxicity has not been paid much attention; thus, the literature is very scarce. The mechanism by which cisplatin induces damage to epithelial cells, neurons or glia cells is not known. Inflammation and oxidative stress involving NF-κB and TNF-α pathways have been proposed as key players (for more information see [118]). However, although enteric neurons have control over the intestinal movement [119], as shown in Hirschsprung disease in humans where loss of intrinsic enteric nervous system results in reduced or absent gut motility, other factors are also important for normal gut motility. Interestingly, cisplatin can have long-term effects in the gastrointestinal tract also in mice (Figure 3).

### 4.3. Gastrointestinal Toxicity Can Impair Kidney and Brain Function and Vice Versa

Motility in the gastrointestinal tract is regulated by the autonomic nervous system composed of extrinsic (i.e., parasympathetic, vagal nerve (the rest-and-digest), sympathetic (fight-or-flight)) and intrinsic enteric nervous systems (ENS) [119,120,121]. The primary regulator of gut motility is intrinsic ENS, followed by extrinsic ENS (parasympathetic, symphatetic) and the central nerve system. However, the gut microbiota, immune system and gut secretions also interact and modulate gut motility [122]. The gut microbiota can affect intestinal transit by modulating the anatomy of the adult ENS (in a serotonin (5-HT)-dependent fashion) [123] and activity of gut-extrinsic sympathetic neurons [124].

In addition, gut barrier dysfunction (i.e., leaky gut) is associated with various kidney disorders. Recent animal studies have demonstrated a direct link between gut inflammation and structural alterations in the kidneys [125], suggesting that persistent gastrointestinal problems of cisplatin treated patients could be involved in the pathogenesis of long-term kidney pathology. Interestingly, renal complications develop in up to 23% of patients with inflammatory bowel disease [125]. On the other hand, impaired kidney function may contribute to long-term gastrointestinal problems in cancer survivors (uremia, cytokines, etc.) [126]. Furthermore, recent studies have demonstrated multiple complex pathways between the gut and the brain [119], linking chemotherapy induced gut–brain axis dysregulation with cognitive impairment, depression and fatigue [127]. The gut microbiota has also been linked with various neurological disorders [128]. In fact, cisplatin causes gut microbiota dysbiosis directly (i.e., cisplatin affects microbiota [129,130]) and indirectly (injury of epithelial cells and inflammation; mucositis [131]), which in the long term can contribute to chronic kidney disease and cognitive impairment [127,132], all of which are frequent complications of cisplatin therapy in cancer survivors. To date, no work has been undertaken to investigate the effects of cisplatin on the submucosal plexus, smooth muscle cells of the muscle layer in the gut wall, extrinsic nerves (i.e., parasympathetic and sympathetic), or the gut–kidney–brain axis dysfunction.

## 5. Cisplatin Neurotoxicity

Cisplatin causes dose related, cumulative toxic effects on the peripheral and central nervous systems (i.e., peripheral neuropathy, chemo brain). Peripheral neuropathy is characterized by sensory loss, often accompanied by pain, starting in the distal extremities [8,133,134]. Chemobrain is characterized by subtle to moderate cognitive deficits such as a decrease in processing speed, memory, executive functioning, and attention [11]. In humans, 49% to 100% of cisplatin treated patients develop some symptoms of neuropathy [135]. The incidence and the severity increase with higher cumulative dose and longer exposure time to cisplatin. Peripheral neuropathy generally develops after a cumulative dose of 250 to 350 mg/m^2^ [136], usually as mild neuropathy in a few patients. When cumulative dose reaches 350–420 mg/m^2^, neuropathy occurs in up to 50% of patients and after 600 mg/m^2^, neuropathy occurs in almost all patients, however, 30–40% of them develop moderate neuropathy, and 10% of them severe and disabling neuropathy [7,8,135,137].

In mice, serial testing at different cumulative doses of cisplatin showed that neuropathy develops progressively with higher cumulative doses [18,138]. Declines in sensory nerve conduction velocity (SNCV) and sudomotor responses were found from cumulative doses of 10 mg/kg, while reduction in the intensity of the nociceptive response to pinprick painful stimuli occurred at cumulative doses of 40 mg/kg (5 or 10 mg/kg/week up to cumulative doses of 40 mg/kg) [138]. In another study SNCV occur at cumulative dose 16 mg/kg (0.5 mg/kg twice per week up to cumulative doses of 32 mg/kg) [18]. There are many protocols of cisplatin induced mouse neurotoxicity [139]. They differ in the dosage, frequency of administration, cumulative dose and consequently in the severity of neurotoxicity and measured outcomes, i.e., the mortality, intensity and the incidence. The most characterized protocol for cisplatin neurotoxicity in mice is administration of cisplatin in two cycles, where one cycle is composed of daily intraperitoneal injection of cisplatin at a dose of 2.3 mg/kg for 5 days, followed by 5 days of recovery (cumulative dose 23 mg/kg; see Figure 4). This protocol induces structural, functional and molecular changes in the peripheral sensory neurons, dorsal root ganglia (DRG), spinal cord, and the brain. Changes can be observed 3–5 weeks after first cisplatin injection. Mice show altered behavioral responses to thermal and mechanical stimuli and impaired performance in the novel object and place recognition tasks. However, although the induced neuropathy is mild and reversible [46], no study reported how many mice develop peripheral neuropathy (the incidence and severity of neuropathy is dose dependent). It has been recognized that models of mild neuropathy have higher inter-individual differences, which requires a higher number of animals per group [140]. In the literature, we can find cisplatin protocols for peripheral neuropathy with an even lower cumulative dose of cisplatin and/or a shorter time point of testing. Considering that neurotoxicity is dose- and time-dependent, such cisplatin protocols do not induce all characteristics of peripheral neuropathy and need to be taken with caution.

Research on cisplatin toxicity of the central system started recently. Advanced neuroimaging techniques in cancer patients have revealed that chemotherapy causes structural alterations in white and gray matter, alterations in the activation of the fronto-parietal attentional network in cancer patients [141], and changes in structural brain networks [142,143]. Cisplatin can cross the blood–brain barrier and penetrate into the brain in low concentrations [15,133] and causes alteration in various parts of the brain in humans and rodents [144]. Structural abnormalities in cerebral white matter [145,146], reduction in myelin density [147], and cerebral neurogenesis [146], changes in synaptic integrity in the prefrontal cortex [148] and decrease in global functional neuronal connectivity in the brain were found also in mice [147]. Cisplatin induced mitochondrial dysfunction and structural abnormalities in brain synaptosomes in the hippocampus [147]. Mice with higher cumulative dose and longer exposure time to cisplatin developed even more severe impairment of mitochondrial transport and mitochondrial dysfunction [149], showing dose dependent toxicity.

### 5.1. Behavioral Tests and Their Weakness

Various behavioral tests have been used to evaluate mice wellbeing, motor activity behavioral responses to mechanical and thermal stimuli, and cognitive performance. It was consistently reported that this cisplatin protocol (Figure 4) induces changes in mice response to radiant heat-paw, tail immersion, adhesive removal test and the von Frey test. Alterations were interpreted as heat hyperalgesia and mechanical allodynia [150,151,152,153]. The pattern of onset and progression of the heat hyperalgesia was similar to the mechanical allodynia and persisted for up to 5 weeks post treatment [150]. No difference was observed in the open field test, motor coordination or signs of paresis (the rotarod test) [153], cold plate test, locomotor activity, grip strength (muscle strength) [150,151]. However, activity patterns of cisplatin treated mice did alter moderately [153], the exploratory activity and body weight of mice were reduced and recovered after cessation of the cisplatin treatment [151]. It was claimed that this cisplatin protocol does not cause significant deterioration in the general health of mice. However, two independent research groups reported body weight decrease (10% after the first cycle and 17% after the second cycle) and sudden death of a mouse during the study [150,151,153]. 

Why is all this information important? In humans, peripheral neuropathy is characterized by sensory loss and pain. Patients describe a range of predominantly sensory, bilateral symptoms in both hands and feet (i.e., a stocking and glove distribution) such as numbness, tingling, spontaneous pain, and hypersensitivity to mechanical and/or cold stimuli [8,14,133,134]. Loss of cognitive abilities of concentration, attention, learning and memory, and executive functions are characteristics of chemotherapy induced cognitive impairment [154].

The pain, sensory abnormalities and cognitive abilities are difficult to evaluate without verbal communication. In animals, therefore, various behavioral tests are used. However, the behavioral tests have many drawbacks. A major shortcoming is that they are all evoked responses. Mice and rats are prey species and when distressed they will mask their spontaneous behavior, sensations and signs of pain. There are many factors that can influence and confound behavioral tests, for instance, aggression (males are prone to aggression) [155], gender of the experimenter (exposure to male experimenters causes in mice stress that results in stress-induced analgesia) [156], anxiety and/or agitation (caused by over-handling or repeated testing) [153] and health states like kidney injury and visceral pain. We have explained that cisplatin causes pica and dose related injuries and inflammation along the gastrointestinal tract, all of which result in visceral pain. Mice suffering from visceral pain of lower abdomen respond to mechanical and thermal stimulation of the hind-paw or tail in the same manner as mice with peripheral neuropathy [157]. It was demonstrated that inflammation in the gastrointestinal tract activates satellite glial cells in DRG and cause excitation of those DRG neurons that innervate particular parts of the gut [158,159]. Major sensory nerves that arise from the L4–L6 DRG neurons innervate the colon [120,160]. These DRG neurons are examined in cisplatin neuropathy studies. Accordingly, visceral pain can be mistakenly diagnosed as peripheral neuropathy. In addition, repeated cisplatin treatment worsens gut toxicity and induces delayed pica, thus, conditioned place preference test used to test the analgesics for cisplatin neuropathy [161] might also be mistakenly interpreted as peripheral neuropathy treatment. None of the neurotoxicity studies evaluated kidney or gut damage.

All the above demonstrates the need for understanding the characteristics and the complexity of cisplatin mouse models to correctly design and interpret the study outcomes. It also demonstrates that outcomes of behavioral tests alone are not sufficient to characterize the model or to evaluate the role of a particular gene or therapeutic agent in the model.

To evaluate and confirm cisplatin induced neuropathy in rodents it is recommended to use behavioral, electrophysiological and histological tests [162]. Electrophysiological tests have limitations. The most significant drawback of the conduction velocity changes is that nerve conduction velocities do not correlate with symptoms [162]. In addition, results of the electrophysiologic tests can vary among studies and even within the laboratory, due to many factors including mice’s body temperature during the recording (the tests are done under anesthesia) [162]. The most reliable is histological assessment, light and electron microscopy. A relevant indicator of small-diameter sensory nerve fiber status in neurotoxicity studies is analysis of intra-epidermal nerve fibers, a method also used for evaluation of peripheral neuropathy in patients, which has yet to become a routine end point in nonclinical safety testing [163]. However, we also need to perform autopsies and analyze all vital organs, particularly the gut and the kidney to evaluate the severity of the injury and inflammation and correctly report and interpret the study outcomes.

### 5.2. Cisplatin Mechanisms

Cisplatin exerts its antitumor activity by binding to guanine and adenine residues, forming cisplatin–DNA adducts that bend and unwind the DNA helix (i.e., distorting its structure by intra- and inter-strand DNA cross-linkage), thus interfering with DNA replication and/or transcription which results in DNA damage, induction of cell cycle arrest, inhibition of DNA synthesis and repair, senescence or cell death (by activating necrotic and apoptotic pathways). While these effects of cisplatin on cancer cells are desired, the same process in normal tissue causes varying degrees of toxicity [5,129,130]. In dividing stem or progenitor cells (myelotoxicity, gut stem cells, etc) cisplatin induces different types of cell death, while in non-dividing cells transcription and translation are more affected leading to senescence, degeneration or dysfunction. Similar to the effect in kidneys (tubular cells), cisplatin was reported to cause DNA damage, activation of apoptotic pathways like p53 activation, Bax translocation, mitochondrial cytochrome C release, activation of caspase-3 and caspase-9 and cell death also in DRG sensory neurons [170].

DRC neurons are non-dividing cells that need a high level of active transcription to sustain their large size, high metabolism, and long axons [171]. Repeated cisplatin administration results in accumulation of cisplatin–DNA adducts in DRG neurons, which is subsequently removed and repaired by nucleotide excision repair (NER) [18]. NER is one of the major DNA repair pathways particularly relevant for cisplatin–DNA adduct repair. Serial testing with increasing cumulative doses of cisplatin showed that mice with NER dysfunctions accumulated higher numbers of cisplatin–DNA adducts in DRG neurons and developed higher severity of peripheral neuropathy [18].

In 2011, Podratz and coworkers demonstrated that cisplatin binds not only nuclear DNA but also mitochondrial DNA (mtDNA), both with the same binding affinity [170]. However, in contrast to nuclear DNA, in mitochondrial DNA isplatin–DNA adducts inhibited mtDNA replication and transcription of mitochondrial genes which resulted in mitochondrial vacuolization and degradation. It was proposed that mitochondrial dysfunction is very likely the consequence of reduced repair of cisplatin adducts in mtDNA, particularly NER [170]. Until recently it was believed that mitochondria do not possess NER. However, extensive investigation in the last decade has shown that mitochondrial DNA repair is very diverse and complex. Mitochondria have an NER mechanism, but it differs from the nuclear one. The proteins that participate in the NER mechanism are imported into the mitochondria in response to oxidative stress [172]. Thus, it is possible, that the NER mechanism is indeed involved in mitochondrial dysfunction (not only in DRG neurons but also in other tissues with high amounts of mitochondria like proximal tubular cells [19] and the brain [41]). However, the contribution of NER in mitochondrial dysfunction remains to be determined.

Loss of mitochondrial number was found in axons of the sensory nerve (tibial) [164], while in the DRG neurons and the brain, mostly alterations in mitochondrial morphology [170] and gene expression were observed [146,164], which suggests that mitochondria were injured but still able to cope and maintain basal functions. However, with higher cumulative dose and longer exposure time to cisplatin more severe impairment of mitochondrial transport and function occurs [149], showing dose dependent toxicity. Interestingly, mitochondrial damage has been investigated and linked with cisplatin toxicity in renal cells of proximal tubules already in the 1980s [63]. Nevertheless, the main cause of cisplatin toxicity remains unknown. We must recall that cisplatin can affect a wide variety of molecules and mechanisms in the cell, it binds not only to DNA but also to various proteins and affects their numerous functions, influences the transport in the cells, etc., [173] (Figure 5).

## 6. Cisplatin Ototoxicity

Cisplatin ototoxicity is a common cisplatin side effect. Cisplatin treated cancer patients experience progressive, bilateral, primarily high-frequency sensorineural hearing loss. Ototoxicity is dose-dependent cisplatin side effect which can start at doses from 60 mg/m^2^/cycle and affects approximately 62% patients. However, in high dose treatment schedules (150–225 mg/m^2^/cycle) up to 100% of patients can be affected [174]. It is reported that 40–80% of adults [16] and 60% of children develop permanent hearing loss [175]. A recent study reported that young children (<5 years) are more susceptible than older children (>5 years). Since young children develop hearing loss at lower cumulative dose and early during cisplatin therapy, audiological monitoring is recommended at each cisplatin cycle [176]. The exact mechanism responsible for hearing loss is not fully understood [174,177,178], but data suggest that cisplatin directly stimulates the production of cytokines leading to inflammation, oxidative stress, endoplasmic reticulum stress and, finally, to various forms of cell death [179]. Currently, there is no treatment to reduce cisplatin ototoxicity [174,177,178]. However, sodium thiosulfate, a thiol-containing antioxidant, has shown promising results in a phase III clinical trial [180].

In the past, a wide variety of cisplatin protocols has been used to model ototoxicity. Most frequently a single, high dose of cisplatin has been used and effects were evaluated a few days later (due to high mortality rate, similarly to nephrotoxicity studies) [178]. Repeated administration of low dosage was also used, but frequently resulted in high mortality or inconsistent and small changes in hearing sensitivity [48]. Protocols were recently summarized and can be found elsewhere [178].

Mouse model of ototoxicity is included in this review mostly as an example of an animal study that aimed to establish a clinically relevant and reproducible mouse model [16,48]. Specifically, it is the first study that reported extensive supportive care for mice during cisplatin treatment. The study [16,48] is summarized with hope that supportive care becomes a part of every cisplatin protocol in animal studies.

To get clinically relevant model of cisplatin ototoxicity, Cunningham and coworkers [16] used an already-established cisplatin protocol. However, to optimize the protocol, firstly a pharmacokinetic study was done to get information on cisplatin distribution and elimination rates from various tissues (kidney, liver, inner ear, brain and heart) before and after each cycle (protocol composed of three cycles of a daily ip injection of cisplatin at a dose 3.5 mg/kg for 4 days, followed by 10 days of recovery; cumulative dose 42 mg/kg) [16]. An auditory function and the three doses were evaluated after each cycle and finally the protocol with clinically relevant and reproducible hearing loss with the lowest suffering of the animals was established [48]. Briefly, CBA/CaJ male and female mice were used and treated with above stated cisplatin protocol but three different doses of cisplatin (2.5, 3.0, and 3.5 mg/kg, cumulative dose 30, 36 and 42 mg/kg, respectively) were evaluated. After the second cycle, minimal hearing loss was observed (at 3.5 mg/kg/day) but without significant threshold shifts across frequencies [16]. Mice developed a dose-dependent loss of cochlear outer hair cell function (distortion product otoacoustic emissions; DPOAEs) and hearing sensitivity (auditory brainstem response; ABR). No significant difference was found between male and female mice. A cisplatin dose of 3.0 mg/kg/day showed better health state of mice than 3.5 mg/kg/day but similarly robust hearing loss across all frequencies, most severe at the high frequencies [48]. This dose (3.0 mg/kg) was thus selected for further characterization of cochleotoxicity and vestibulotoxicity. Assessment of auditory function follows 42 days after the first cisplatin injection. It was found that after cessation of cisplatin administration hearing loss in mice even progresses over time [16,48], similar to cisplatin ototoxicity in humans [174].

### Hydration and Supportive Care in Cisplatin Protocols


From the first day of the study, all cisplatin treated mice received intensive supportive care (twice daily). Supportive care was composed of hydration (1 mL of 0.9%NaCl and 1ml of Normasol injected subcutenously) and supplemental nutrition (0.3 mL high calorie liquid supplement, DietGel Recovery cups and pellets on the floor cage). Body weight, overall health, activity and body condition scoring [181] was used to monitor the overall condition of each mouse on daily basis (muscular tone, body fat content, coat maintenance, overall energy level. Using supportive care protocol, all mice in the study survived although their body weight progressively decreased during each cycle and at the end of the experiment reached significant loss of their initial weight (21% at dose 2.5 mg/kg and 27% at dose 3.0, 3.5 mg/kg). The only drawback of this study is that the kidney function and gastrointestinal damage in mice were not examined. Inflammation has significant effects on health and disease. Treatment of inflammation (in the gut and kidney) could improve the mice’s health state. Particularly because during the auditory testing mice need to be anesthetized, and diseased animals are at higher risk of death during the anesthesia. Thus, during anesthesia special care is needed to avoid additional hypothermia, hypoxia, acidosis, and death.

We must recall that mice treated with cisplatin suffer from acute and delayed pica, gastric distension (delay in gastric emptying, stomach filled with bedding), reduced food intake, inflammation in intestine, polyuria (malnutrition, dehydration and electrolyte waste). In addition, mice treated with cisplatin are hypothermic [182]. Mice that are ill and suffer abdominal pain (intestine inflammation, nausea/pica/full stomach, kidney injury) are less active and vital, do not rear/climb up after water and food, and do not care for their nests. Well-structured nests are important for their body temperature maintenance. Supportive care is mandatory, to prevent agonistic death from dehydration, malnutrition and hypothermia. Vitamin C and sodium bicarbonate pretreatments has been show to improve mice’s health and reduce cisplatin nephrotoxicity [182], while dexamethasone, a corticosteroid used in humans and/or ondansetron, a serotonin 5-HT3 receptor antagonist, showed confounding results [106].

It is interesting that in the 1980s the effects of hydration and cisplatin vehicle (Table 5) on nephrotoxicity and tumor burden were tested in cisplatin treated mice and rats. Although both hydration [183] and the vehicle in which cisplatin was dissolved [184,185] markedly reduced mortality and nephrotoxicity, hydration became a routinely used method of nephrotoxicity prevention only in clinics but not in preclinical models. Intravenous hydration using isotonic saline solution significantly reduces cisplatin half-life, urinary cisplatin concentrations and proximal tubule transit time [3,12], which reduces nephrotoxicity and allows higher doses of cisplatin for the cancer treatment. Thus, hydration affects the MTD dose and, consequently, also the therapeutic effect of cisplatin (dose dependent) in preclinical studies (see Section 3.3). In addition, not only the incidence but also the severity of cisplatin toxicity is dose dependent. Based on the cisplatin protocol mice thus can develop (Figure 6): changes in molecular mechanisms without structural damage (process is in the range of the physiological limits and does not affect the clinical picture, although molecular markers can show significant increases; MTD); changes in molecular mechanisms with structural damage (although structural damage is histologically confirmed and clinical signs are present, damage is still in the range where regression and restitution or repair is possible; mild, moderate); clinical signs are present and structural damage is obvious, regression and repair is possible only if properly treated (severe-systemic inflammation) and intoxication (irreversible).

Various factors can affect response to cisplatin treatment such as strain, substrain [44,106], age [105,186], hydration [183], circadian rhythms [23,187,188,189]. However, there is high inter-individual variability also among mice within the same inbred strain (genetically uniform) showing that environmental and phenotypic factors like physical state play important roles in cisplatin toxicity. Since there are many factors that can influence cisplatin effects (therapeutic or toxic) scientists are encouraged to thoroughly report all details in their study and follow the ARRIVE guidelines [190] or the Gold Standard Publication Checklist [191], FELASA recommendations [192,193] and standardized genetic nomenclature of rodents (http://www.informatics.jax.org/nomen/strains.shtml)6^th^ October 2021.

## 7. Cisplatin Distribution and Elimination

To better understand the complexity of cisplatin toxicity in various organs, basic knowledge about cisplatin distribution, elimination and accumulation is briefly summarized. Cisplatin reaches systemic circulation within 10 min after systemic administration (ip, iv) [15,16,194] and within 1 h cisplatin is already distributed in almost all tissues studied (kidney, liver, lung, inner ear, heart and brain), with the highest concentration in the kidney [16]. There is a linear correlation between cisplatin dose (3.75, 7.5 or 15 mg/kg) and cisplatin concentration in the blood or tissues (kidney, liver, tumor, brain and testis) 1 h after ip administration [15]. Free cisplatin eliminates from the blood predominantly by the kidney, much less by biliary [194] or intestinal excretion [184].

It appears that the rate of cisplatin clearance in repeated treatment depends on the dose (cumulative) and the frequency interval (daily vs weekly). Repeated administration of low dose of cisplatin (16 mg/m^2^ or 2.5 mg/kg) did not affect the elimination rates of cisplatin until the fifth cycle (ip; five cycles with 3-week intervals between each cycle). After the fifth cycle elimination of cisplatin significantly decreased (cumulative dose reached 12.5 mg/kg) [195]. In contrast, repeated administration of higher doses of cisplatin (5 mg/kg iv; three cycles with 3 weeks between each cycle) resulted in decreased renal clearance and increased accumulation of cisplatin in the kidney by each cycle (cumulative dose at the second cycle reached 10 mg/kg) [196], suggesting a longer elimination half-life of cisplatin and an impaired elimination/detoxification mechanisms when reaching critical levels of cisplatin (Table 3). A similar situation occurred in the case of cisplatin protocol for ototoxicity (three cycles of 14 mg/kg (3.5 mg/kg/daily) with 10-day intervals between each cycle). After each cycle the elimination of cisplatin decreased, resulting in gradual retention of cisplatin in tissues. After the third cycle (42 days after the start) cisplatin in all examined tissues reached levels twofold higher than after the first cycle. The highest concentration of cisplatin was detected in the liver, followed by spleen, femur, kidney, inner ear, lung, heart, skeletal muscle, small intestine and brain. Two months later (60 days recovery) marked decline was observed in all tissues except femur and inner ear. However, in all examined tissues cisplatin was still present at the detected levels [16].

While elimination of cisplatin from the blood is very rapid (mostly within 1 h), elimination from the tissues is a longer process lasting weeks or even years. In tissues cisplatin accumulates in all cell compartments, the mitochondria, nucleus, cytoplasm, microsomes [195,197]. In general, the larger decline of cisplatin concentration in tissues occurs within the first 24 h [15,16], followed by slower elimination rates during the first 30 days [15,79,195] reaching an almost steady state 3 months after a single nephrotoxic dose of cisplatin [195]. Recently it was found that elimination rates from the inner ear are much lower than in other organs. Cisplatin retains in the inner ear for months in both mice and humans (at least 18 months after patients last cycle) [16]. It was also found that the highest levels of cisplatin in the inner ear accumulate in the stria vascularis (the region of the inner ear that maintains the ionic composition of endolymph), while cisplatin accumulation in mechanosensory hair cells is more limited. Similar cisplatin distribution was also found in humans. Long-term retention of cisplatin was associated with progressive hearing loss in mice [16].

Cisplatin retention in the tissues can be evaluated also by detecting cisplatin-DNA adducts, a method usually used in the nervous system (see section neurotoxicity). However, cisplatin–DNA adducts can be found also in other tissues (kidney, liver, testis and brain). Nevertheless, the formation of cisplatin–DNA adducts is a slower process; depending on the tissue it can take up to 4 h or more [15,198]. After single dose of cisplatin (7.5 mg/kg) the highest levels were observed in the kidney cortex, particularly tubules. The levels persisted for 24 h (liver, kidney), followed by a slow decline, while in other tissues (tumor, testis) decline was observed within first 12 h. Formation of cisplatin–DNA adducts was dose dependent with large inter-individual variations, particularly for kidney and tumor [15]. Cisplatin–DNA adducts can be detected in various tissues in patients treated with cisplatin for many months after therapy [198].

## 8. Discussion

As shown in the paper, there are many similarities between mice and humans. Mice develop all cisplatin side effects in a dose- and time-dependent manner. Just as humans, mice also develop cisplatin side effects of varying severity from mild to multi-organ failure, each pathology with its own time course and pathophysiological response or molecular signature. Despite all the similarities, there is an apparent gap between the results in animal models and human clinical trials.

As described, there are many drawbacks in the currently used cisplatin protocols. Besides a wide variability in protocols [44], most of cisplatin protocols have no similarities to the treatment schedules used in cancer patients. In humans, cisplatin is given in cycles with extensive hydration and supportive care to provide the highest possible dose of cisplatin to improve the success of therapy, while in tumor bearing mice a wide variety of cisplatin protocols with no hydration or supportive care are used. In mice, cisplatin treatment ranges from a single to repeated (multiple) administration, where cumulative doses range from sub-therapeutic to lethal doses or even higher (see Table 1 and Appendix A). To evaluate potential beneficial effects of therapy or toxicity, in mice studies, most frequently only the size or the volume of the tumor is used as a measure of successful treatment and the body weight is used as a marker of systemic toxicity. No examination of gut toxicity, myelotoxicity or neurotoxicity is performed. Rarely, a few blood parameters are examined. Body condition of the animals and mortality rate are rarely reported and necropsy and histology of all vital organs are rarely performed (Appendix A). Importantly, cisplatin protocol, hydration and supportive care all together affect not only the MTD or lethal dose but also the therapeutic dose of cisplatin and its side effects (dose-dependent). Higher doses of cisplatin result in higher cisplatin tissue retention (see section cisplatin distribution and elimination).

As demonstrated in the article, mice respond to cisplatin therapy in a similar way to humans. Importantly, mouse response to cisplatin is highly dependent on cisplatin protocols. Thus, we can say that we get what we design. If we want to understand molecular mechanisms and find therapeutic agents that would have a potential benefit in clinics, we need to use similar cisplatin treatment protocols as are used in cancer patients.

In this paper, only the most characterized cisplatin protocols were presented together with weaknesses that need to be improved in future studies. An example of hydration and supportive care in repeated cisplatin protocol is summarized with the hope that in the future hydration and supportive care become a part of cisplatin protocols. The use of the same cisplatin protocol by various research groups around the world could help evaluate, optimize and validate particular cisplatin protocols. Investigating cisplatin effects in all organs of a currently established model and gaining insight into complex cisplatin toxicology would help understand the underlying mechanisms of cisplatin toxicity in a time-dependent manner. It would enable the use of optimal markers of a certain toxicity at a given time period/point in the development of the toxicity. Optimized and validated models can then be used to test potential treatment strategies for cisplatin toxicity. However, first optimization with hydration and supportive care is needed. This may affect the dose adjustment in cisplatin protocols. Then protocols need to be tested and optimized in tumor-bearing animals.

Research on mice enables systematic and controlled investigation of complex mechanisms involved in the development of cisplatin therapeutic or toxic effects. In addition, it enables investigation of pathogenesis of cisplatin toxicity in a time- and dose-dependent manner. However, it is important that we change our approach to animal studies and start to treat animals in research as patients and not as a tool. Otherwise we must ask ourselves *“what have we chosen to ignore in this model, and at what cost?”*[199].

## Figures and Tables

**Figure 1 biomedicines-09-01406-f001:**
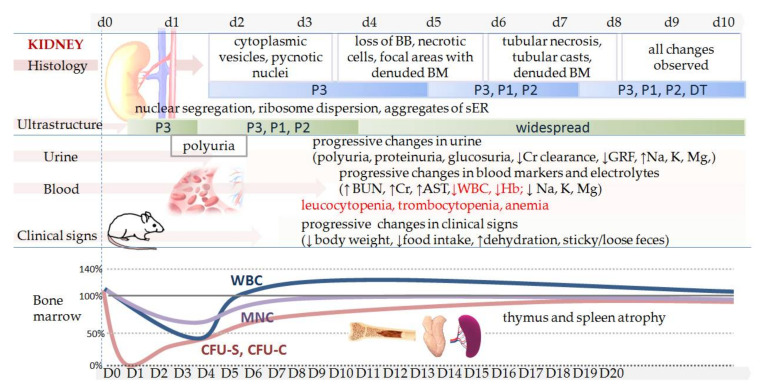
**S**chematic presentation of clinical signs and kidney function in mice after a nephrotoxic dose of cisplatin. First, two days after a single high dose of cisplatin (10–13 mg/kg; ip), minimal structural changes in the proximal tubules (P3) can be detected (i.e., mitochondria alterations, focal loss of the microvillus brush border, pycnotic nuclei, increased cytoplasmic vesicles) [63,64]. More obvious changes such as loss of the brush border or necrotic cells sloughing into the tubular lumen are usually seen 3–4 days after injection and changes are located in all parts of the proximal tubules (P1–3)[63,64,65]. Depending on the dose, increased BUN/Cr are usually observed 3–7 days after cisplatin injection [66,67,68,69], and if nephrotoxicity is reversible, BUN/Cr return to the baseline levels within 14 days [70]. In such cases, the first signs of structural regeneration can be observed 7 days after cisplatin injection [64,71]. A single high dose of cisplatin (B6D2F1: 8 mg/kg, 10 mg/kg, 12 mg/kg, 14 mg/kg; ip) induces dose-dependent weight loss (11–26%), reticulocytopenia with the lowest levels of body weight and reticulocytes observed 6 days after cisplatin injection. Necrosis in kidney tubular cells can be seen up to 10–22 days post-treatment [72]. When a lethal dose is used, death may occur within 10 days [73] and the time course of AKI development or mortality can occur slightly faster, but still 1–2 days after cisplatin injection. Cisplatin (F1 CBAxC57BL, 12 mg/kg, ip) induces lymphocytopenia, thrombocytopenia and anemia. Cisplatin exhibits cytotoxicity to spleen (CFU-S), granulocyte–macrophage (CFU-C) colony-forming units and mononuclear cells (MNC) in bone marrow and white blood cells (WBC) (adapted and modified from Nowrousian et al. [74]) Legend: P1–3 denotes kidney proximal tubules parts 1–3, DT—distal tubules; BB—brush border; BM—basal membrane; BUN- blood urea nitrogen; Cr—serum creatinine; AST—aspartate aminotransferase; WBC—white blood cells; Hb—hemoglobin; GFR—glomerular filtration rate; CFU—colony-forming unit; MNC—mononuclear cells; ER—endoplasmic reticulum.

**Figure 2 biomedicines-09-01406-f002:**
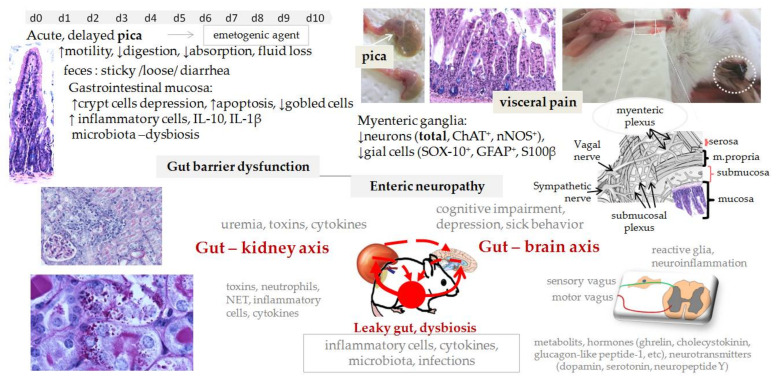
Cisplatin causes acute and chronic effects in the gastrointestinal tract. A single injection of cisplatin causes pica, a rodent-specific behavior of nausea, which reflects as a progressive reduction in food intake, increase in non-nutritive material intake (for instance bedding) and decreased gastric motility [107]. As a result the stomach is full of bedding and markedly enlarged/distended (white arrow) [114]. Reduction in food (68%) and water intake (45%) and an increase in stomach content (threefold) is evident from day 2 on (C57BL/6; 6 mg/kg ip) [114]. First morphological changes in the small intestinal mucosa (i.e., apoptosis, necrosis, decreased number of goblet cells, shortened villi and inflammatory cell infiltration) can be seen 1 day after a single cisplatin injection (B6D2F1: 8 mg/kg, 10 mg/kg, 12 mg/kg, 14 mg/kg; ip; d1,3,6,10,14) followed by reduced mucosal digestive function (depletion in maltase, sucrose, disaccharidase activity and reduced absorption) [108,115]. Depression in crypt cell production is already evident 2h after cisplatin and is maximal between 12 and 24 h post-treatment (CBA: 10 mg/kg, ip). Cisplatin causes lesions also in the colon mucosa, however, they appear later and are less severe [72]. The severity of gastrointestinal damage and mucosal dysfunction is dose-dependent and can persist up to 10 days after a single sub-lethal dose of cisplatin (B6D2F1: 8 mg/kg, 10 mg/kg, 12 mg/kg, 14 mg/kg; ip; d1,3,6,10,14) [72]. Mucosal recovery is slow, first signs of recovery can be observed 7 days post-treatment [72]. Repeated cisplatin administration (C57BL/6; 4 mg/kg/week for 4 weeks, ip; ↓20% BW) besides gut lesions (↑IL-1β and IL-10) also causes delayed pica, [55] and alterations in the ENS seen as loss of neurons in the myenteric ganglia of mouse gastric fundus (total and nNOS^+^) [56] and colon (neurons (total, ChAT^+^, nNOS^+^) and gial cells (SOX-10^+^, GFAP^+^, S100β^+^) [55]. Circulation and the nervous system are the main pathways for communication between the gut, the kidney and the brain in health or disease (the brain–gut–kidney axis). Legend: BW—body weight; ChAT—choline acetyltransferase; ENS—enteric nerve system; GFAP—glial fibrillary acidic protein; NET—neutrophil extracellular traps; nNOS—neuronal nitric oxide synthase; ip—intraperitonealy.

**Figure 3 biomedicines-09-01406-f003:**
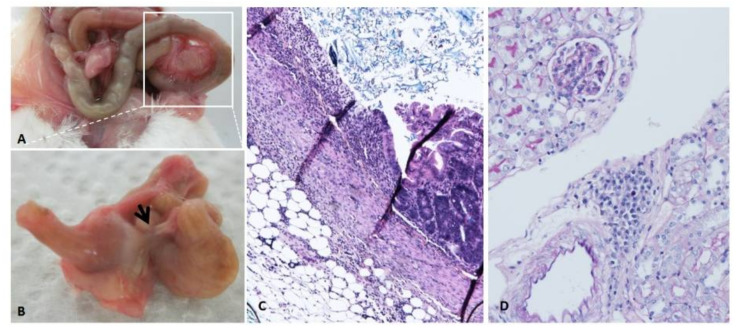
Cisplatin can have long-term effects in the gastrointestinal tract (**A**). A case of penetrating ulcer (**B**, arrow and **C**) in a mouse that survived a single lethal dose of cisplatin (17 mg/kg). Three months after cisplatin recovery, body weight started to decrease, and the mouse was killed and autopsy performed. Inflammatory cells found in the kidney (**D**).

**Figure 4 biomedicines-09-01406-f004:**
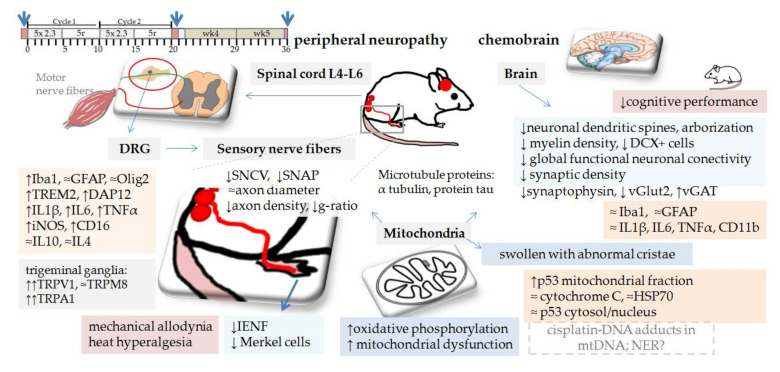
Cisplatin neurotoxicity. In mice, two cycles of cisplatin (2.3 mg/kg/daily for 5 days followed by 5 days recovery; 5d+5r/5d+5r; cumulative dose 23 mg/kg) resulted in reduced density of intraepidermal nerve fibers (IENF) (wk3, and wk5) [152,164] and epidermal Merkel cells [152] in the mouse plantar footpad. Merkel cells, mechanosensory cells actively involved in touch reception (tactile sensation), [165,166,167] are proposed to underlie sensory dysfunction in diabetic patients and animals [168]. In sensory nerves (sciatic, caudal, tibial) mild hypomyelination with few degenerating axons (reduced density of myelinated fibers without alterations in axon diameter) can be observed together with a slight decrease in the sensory nerve conduction velocity (SNCV; indication of demyelination) and the sensory nerve action potential (SNAP) [153]. In sensory neurons (trigeminal ganglia) cisplatin activated the transient receptor potential (TRP) channels (TRPA1, TRPV1) [151], a non-selective cation channels involved in chemical and thermal evoked pain sensation [169]. In the spinal cord (L4-L6) cisplatin activated microglia (Iba1), induced pro-inflammatory cytokines (IL-1β, IL-6, TNFα, iNOS, CD16, a marker of pro-inflammatory microglia (wk3) and increased protein levels of triggering receptor expressed on myeloid cells 2 (TREM2) and DNAX activating protein of 12 kDa (DAP12) (wk3) [152]. TREM2/DNAX is a receptor complex predominantly expressed on microglia in the central nervous system associated with neurodegenerative diseases and inflammatory response of microglia [152]. Cisplatin induced structural abnormalities in cerebral white matter (loss of neuronal dendritic spines and arborizations) [145,146] and reduced myelin density in the cingulated cortex [147]. It also [145] decreased cerebral neurogenesis (DCX^+^ cells) [146] but did not cause inflammation (IL1β, IL6, TNFα, GFAP, CD11b) [146] or microglia (Iba1^−^, GFAP^−^) activation [145]. However, decreased synaptic integrity (synaptophysin, vGlut2, vGAT) in the prefrontal cortex [148] and global functional neuronal connectivity in the mouse brain was found (fMRI) [147]. Cisplatin induced mitochondrial dysfunction and structural abnormalities in brain synaptosomes [147]. Mice treated with three cycles of cisplatin (protocol 2.3 mg/kg 5d + 5r/5d + 5r/5d + 5r; cumulative dose 34.5mg/kg) developed more severe impairment of mitochondrial transport and mitochondrial dysfunction in the hippocampus [149] (43% decrease in cytochrome C activity, ATP production, 96% increase in ROS, 29% decrease in mitochondrial membrane potential, impaired mitochondrial transport, reduced α-tubulin acetylation in the hippocampus, decrease in dendritic spine and synaptic density (vGlut1 and PSD95) [149]. Legend: DAP12—DNAX activating protein of 12 kDa; DRG—dorsal root ganglia; GFAP—glial fibrillary acidic protein; IENF—intraepidermal nerve fibers; IL—interleukine; Iba1—ionized calcium-binding adaptor molecule 1; iNOS—inducible nitric oxide synthase; L4-L6—lumbal vertebra; mtDNA—mitochondrial DNA; NER - nucleotide excision repair; Olig-2—oligodendrocyte lineage gene 2; ROS—reactive oxidative species; SNAP—sensory nerve action potential; SNCV—sensory nerve conduction velocity; TNFα—tumor necrosis factor alpha; TRP—transient receptor potential channels (TRPA1, TRPV1); TREM2—triggering receptor expressed on myeloid cells 2; vGlut2—vesicular glutamate transporter 2; vGAT—vesicular GABA transporter.

**Figure 5 biomedicines-09-01406-f005:**
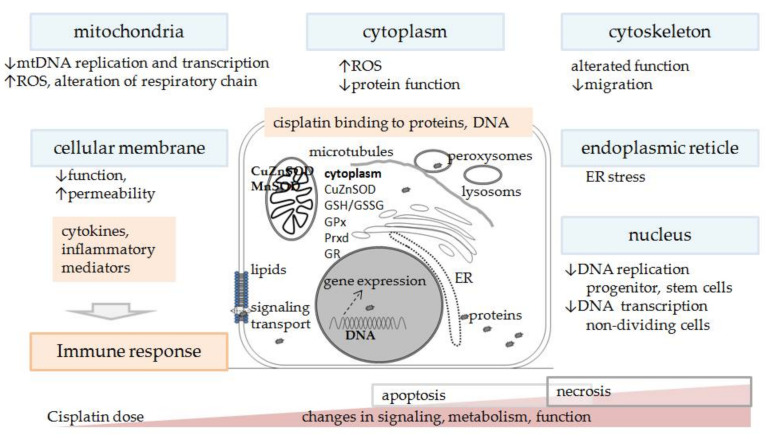
Schematic presentation of cisplatin toxicity in non-tumor cells in the body. Extent and intensity of oxidative stress, changes in signaling, metabolism, function, intensity of inflammation, activation of certain immune cell types, inflammatory and molecular crosstalk and response, type of cell death, etc., depend on cisplatin dose (single or cumulative) and severity of toxicity. ER—endoplasmic reticulum; mtDNA—mitochondrial DNA; ROS—reactive oxygen species.

**Figure 6 biomedicines-09-01406-f006:**
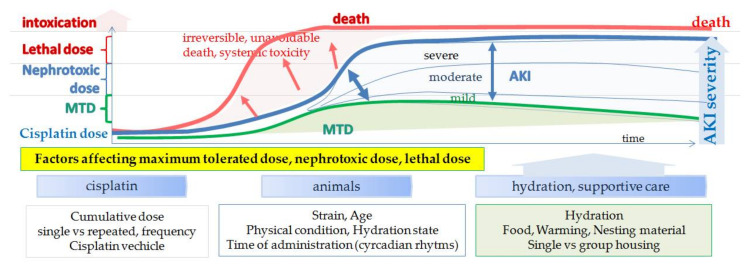
Dose-dependent toxicity of cisplatin and factors affecting maximum tolerated dose (MTD), nephrotoxic and lethal dose.

**Table 1 biomedicines-09-01406-t001:** Examples of mouse cisplatin toxicity protocols used in studies published from April 2020 to February 2021.

Cisplatin Toxicity (100%)	Cisplatin Protocols	Endpoint
Nephrotoxicity (57.1%)	8, 10, 15, 20, 25, 30, 40 mg/kg single ip (most frequently used 20 mg/kg single ip)	d3–d4
Gastrointestinal toxicity (3.6%)	20 mg/kg single ip	d3
Ototoxicity (10.7%)	30 mg/kg single ip (FVB; hydration 1ml 2xdaily; 50% mortality) 3 cycles: 3 × (3–3.5 mg/kg/daily for 4 days followed by 10 days recovery) Cumulative dose = 36 or 42 mg/kg	d21 d42
Neurotoxicity (10.7%)	2 cycles: 2 × (2.3 mg/kg/daily for 5 days followed by 5 days recovery) Cumulative dose = 23 mg/kg	d15, d30, d65
Gonadotoxicity (10.7%)	3 × 5 mg/kg ip 5 × 3mg/kg ip	d4 d14
Muscle (3.6%)	4 × 3 mg/kg/daily	d4
Anemia (3.6%)	4 × 7mg/kg/week ip	2 months

**Table 2 biomedicines-09-01406-t002:** Examples of cisplatin regimes used in the clinics and incidence of AKI complications.

Cisplatin Clinical Dose in Humans ^1^ (iv)		AKI Incidence, Severity	ref	MED??
50–75 mg/m^2^#	1.35–2.03 mg/kg	25–33%, mild-moderate	[62]	16.7–25.0 mg/kg
15–20 mg/m^2^ daily for 5 days#	0.41–0.54 mg/kg	50–75%, mild-moderate	[62]	5.0–6.7 mg/kg
100 mg/m^2^#	2.7 mg/kg	severe to irreversible	[62]	33.4 mg/kg
75 mg/m^2^ every 3 weeks up to 6 cycles *	2.03 mg/kg	53%, mild-moderate	[89]	25 mg/kg
100 mg/m^2^ with concurrent radiation **	2.7 mg/kg **	47–60% of patients discontinued therapy	[90]	33.4 mg/kg
80 mg/m^2^ 1 h iv infusion	2.2 mg/kg	H#	[91]	27 mg/kg

^1^ dose of cisplatin in humans measured as mg per skin area (mg/m^2^) was translated in mg/kg using the correction factor for human body weight of 60 kg and the body surface area 1.62 m^2^ (K_m_ = 37) [92]. #data from 1978 when supportive care measures were not established; * therapy cisplatin/docetaxel (lung cancer); cisplatin 2 h infusion every 3 weeks, antiemetic prophylaxis, hydration with up to 3000 mL of normal saline; cumulative dose = 340 mg/m^2^ [89]; ** 2–3 cycles every 3 weeks; doses of cisplatin for subsequent cycles were adjusted at the discretion of the physician; (squamous cell cancer of the head and neck); cisplatin 2 h infusion diluted in 1 L of 0.9% saline and 1–2 h hydration with 1 L of saline pre and post cisplatin infusion; antiemetic premedication (dexamethasone, 5-HT_3_ antagonist, neurokinin-1 receptor antagonist) [90]; H#—the highest dose recommended as a single administration[91]. MED—mouse equivalent dose, needs to be treated with caution (see warning in Section 3.3). Calculation was done according to guide for dose conversion using correction factor for 20 g mouse (K_m_ = 0.081) [92].

**Table 3 biomedicines-09-01406-t003:** Cisplatin acute toxicity and single or repeated maximum tolerated dose given intraperitonealy varies among mouse strains.

Strain, Sex, Age	Single Dose LD100	Endpoint	Ref.
BALB/c, female, (N = 8)	14.5mg/kg; ip	d7	[69,103]
C57BL/6, male, 11–15wk; (N = 5)	15 mg/kg; ip	d10	[104]
CBA; female, 24 months, (N = 3)	16 mg/kg; ip	d7	[105]
	**Single MTD**		
BALB/c, female, 8–10 wk; (N = 3)	6 mg/kg; ip	d10	[106]
C57BL/6J, female, 8–10 wk; (N = 3)	6 mg/kg; ip	d10	[106]
	**Repeated MTD**		
C57BL/6J, female, 8–10 wk; (N = 3)	3 × 4 mg/kg; ip	d21	[106]

Repeated administration: once mice had recovered to 100% of their starting weight or a clinical score of 0, a second MDT was given (d0, d8, d16). MTD is defined as a dose as high as possible that causes no unacceptable toxicity such as no clinical evidence of toxicity, no reduction in mean body weight >10% to 15% and, no mortality [106]. Legend: N: number of animals; ip: intraperitonealy; iv: intravenously; LD: lethal dose; LD100: dose of cisplatin that results in 100% mortality in animals (without hydration or supportive care); d: day, MTD: maximum tolerated dose.

**Table 4 biomedicines-09-01406-t004:** An example of an incidence and severity of cisplatin acute toxicities in cancer patients.

Severity (Grade)	Any (1–4)	Severe (3–4)		Any (1–4)	Severe (3–4)
Nausea	90.7%	23.6%	Anaemia	76.7%	2.3%
Vomiting	58.1%	14%	Leukopenia	83.7%	44.2%
Diarrhea	65.1%	18.6%	Neutropenia	72.1%	55.8%
Constipation	27.9%	0%	Thrombocytopenia	32.6%	9.3%
Stomatitis	55.8%	9.3%	Creatinine	55.8%	2.3%
Neurosensory	53.5%	2.3%	Infection	41.9%	25.6%
Fatigue	81.4%	20.1%	Fever	23.6%	0%
Weight loss	41.9%	2.3%			

Cisplatin 75 mg/m^2^ every 3 weeks up to 6 cycles or until cessation (cumulative = 340 mg/m^2^). Therapy cisplatin/docetacel; cisplatin 2h infusion every 3 weeks, antiemetic prophylaxis, pre and post cisplatin hydration with up to 3000 mL of normal saline [89].

**Table 5 biomedicines-09-01406-t005:** Effect of cisplatin vehicle on cisplatin toxicity/mortality [184,185].

Cisplatin Vehicle	LD50
distilled water	10.8 ± 1.0 mg/kg
0.9% NaCl	15.3 ± 1.6 mg/kg
4.5% NaCl	24.5 ± 0.7 mg/kg

LD50—dose of cisplatin that results in 50% mortality in animals.

## Data Availability

Not applicable.

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
