# Peer review of "Cisplatin Mouse Models: Treatment, Toxicity and Translatability"

_biomedicines, 2021, doi:10.3390/biomedicines9101406_

Round 1

Reviewer 1 Report

This review summarizes a difficult topic. I really appreciate the critical view of the author and the underlining of the problematic issues. The figures nicely recapitulate the timelines of cisplatin induced toxicities.

General

- All the abbreviations should be spelled off. In addition, figure legends should report the full name of the proteins reported in the figures.

- There are some spelling and grammatical errors (i.e. line 578: ototoxocity = ototoxicity)

Introduction section

“However, in contrast to cisplatin treated cancer patients, in animal studies there is usually one cisplatin toxicity under investigation, while other side effects of cisplatin are mostly neglected/ignored.” The sentence should be rewritten in order to have clear which toxicity is usually under investigation.

Paragraph 4.2.

The authors should include also inflammation among the mechanism underpinning cisplatin-induced gastrointestinal toxicity (i.e. NFkB and TNF-α are known to be involved htps://doi.org/10.1016/j.ejphar.2018.03.009). 

Lines 327-328: at least a reference is needed.

Cisplatin ototoxicity paragraph.

This paragraph should be implemented. Even if the author stated that cisplatin ototoxicity was mostly included as an example, I think, it should be better highlight some important messages:

- the need for audiological monitoring at each cisplatin cycle (DOI: 10.1002/cncr.33848)

- Currently, there is no treatment to reduce cisplatin ototoxicity, but an antioxidant, sodium thiosulfate showed promising results in a phase III clinical trial (https://doi.org/10.1016/S1470-2045(16)30625-8)

- the exact mechanism responsible for hearing loss is not fully understood, but recent data suggest that inflammation could be the trigger event leading to inner ear cell death through endoplasmic reticulum stress, autophagy, necroptosis, and then intrinsic apoptosis (doi: 10.1016/j.molmed.2019.08.002)

Author Response

I would like to thank the reviewer for the critical review and very helpful suggestions. I have followed the reviewer's instructions and included all suggested improvements.

Reviewer: All the abbreviations should be spelled off. In addition, figure legends should report the full name of the proteins reported in the figures.

Answer: Done. Abbreviations are at the end of the manuscript and below  figure legends.

Reviewer:- There are some spelling and grammatical errors (i.e. line 578: ototoxocity = ototoxicity)

Answer: Corrected.

Reviewer: Introduction section

“However, in contrast to cisplatin treated cancer patients, in animal studies there is usually one cisplatin toxicity under investigation, while other side effects of cisplatin are mostly neglected/ignored.” The sentence should be rewritten in order to have clear which toxicity is usually under investigation.

Answer: Corrected. In the manuscript, the corrected text is marked yellow.

Reviewer: Paragraph 4.2.

The authors should include also inflammation among the mechanism underpinning cisplatin-induced gastrointestinal toxicity (i.e. NFkB and TNF-α are known to be involved htps://doi.org/10.1016/j.ejphar.2018.03.009). 

Answer: Corrected. In the manuscript, the corrected text is marked yellow.

 Reviewer: Lines 327-328: at least a reference is needed.

Answer: The reference has been added.

Reviewer: Cisplatin ototoxicity paragraph.

This paragraph should be implemented. Even if the author stated that cisplatin ototoxicity was mostly included as an example, I think, it should be better highlight some important messages:

- the need for audiological monitoring at each cisplatin cycle (DOI: 10.1002/cncr.33848)

- Currently, there is no treatment to reduce cisplatin ototoxicity, but an antioxidant, sodium thiosulfate showed promising results in a phase III clinical trial (https://doi.org/10.1016/S1470-2045(16)30625-8) Freyer 2017 Lancet

- the exact mechanism responsible for hearing loss is not fully understood, but recent data suggest that inflammation could be the trigger event leading to inner ear cell death through endoplasmic reticulum stress, autophagy, necroptosis, and then intrinsic apoptosis (doi: 10.1016/j.molmed.2019.08.002) Gentilin 2019 CellPress

Answer: Thank you for the very valuable articles, which are now included in the manuscript. Cisplatin ototoxicity is indeed a very important clinical issue and hopefully, the corrected version now highlights its importance.

Reviewer 2 Report

The article composed by Perse for systemic analysis of literatures regarding toxicological and pharmacological response of rodent models while performing cisplatin tests. Although cisplatin toxicology profile has been well documented in human, this article pointed out potential pitfalls of experimental model for future notice. This article deserves for publication since it summarized various data rodent models. The manuscript was well prepared and composed in reader-friendly manner.

Author Response

I would like to thank the reviewer for the critical review and positive comments.